# Progress in Aromatic Polyimide Films for Electronic Applications: Preparation, Structure and Properties

**DOI:** 10.3390/polym14061269

**Published:** 2022-03-21

**Authors:** Ziyu Wu, Jianjun He, Haixia Yang, Shiyong Yang

**Affiliations:** 1Key Laboratory of Science and Technology on High-Tech Polymer Materials, Institute of Chemistry, Chinese Academy of Sciences, Beijing 100190, China; wuziyu@iccas.ac.cn (Z.W.); hejianjun@iccas.ac.cn (J.H.); 2School of Chemical Sciences, University of Chinese Academy of Sciences, Beijing 100049, China

**Keywords:** aromatic polyimides, films, structure and properties, electronic applications

## Abstract

Aromatic polyimides have excellent thermal stability, mechanical strength and toughness, high electric insulating properties, low dielectric constants and dissipation factors, and high radiation and wear resistance, among other properties, and can be processed into a variety of materials, including films, fibers, carbon fiber composites, engineering plastics, foams, porous membranes, coatings, etc. Aromatic polyimide materials have found widespread use in a variety of high-tech domains, including electric insulating, microelectronics and optoelectronics, aerospace and aviation industries, and so on, due to their superior combination characteristics and variable processability. In recent years, there have been many publications on aromatic polyimide materials, including several books available to readers. In this review, the representative progress in aromatic polyimide films for electronic applications, especially in our laboratory, will be described.

## 1. Introduction

Aromatic polyimides (PI) are an important class of heteroaromatic polymers, which have excellent thermal stability, mechanical strength and toughness, high electric insulating properties, low dielectric constants and dissipation factors, and high radiation and wear resistance, among other properties, and can be processed into a variety of materials, including films, fibers, carbon fiber composites, engineering plastics, foams, porous membranes, coatings and varnishes, etc., and have found widespread use in a variety of high-tech domains, including electric insulating, microelectronics and optoelectronics, aerospace and aviation industries, etc. Despite the fact that the first publication on the synthesis of aromatic polyimide was published in 1908 [1], Dupont released the first commercial polyimide material in the late 1960s after developing an effective synthetic pathway to high-molecular-weight polyimides. Even today, however, the most popular polyimide production processes are not totally understood. This is because numerous reaction conditions, such as solvent, moisture, impurity, temperature, and others, have a considerable impact on the polymerization reaction of aromatic dianhydrides with aromatic diamines. The molecular weights of the produced polyimides can be affected by the addition mode of monomers.

In recent years, there have been many publications on polyimide materials [2,3,4,5,6,7,8,9,10,11,12,13,14]. In this review, the representative progress, mainly in our laboratory, in advanced polyimide films, will be described, especially focusing on the structure and properties for electronic applications.

## 2. Aromatic Polyimide Films

Aromatic polyimide films have been widely used in many high-tech fields, including electrical insulation [15,16,17,18,19,20], electronic packaging [21,22], and aerospace [23,24,25,26,27,28,29,30], due to their excellent combination of thermal, mechanical, and electrical properties. Aromatic polyimide films can be made by casting the polyimide precursor solution–polyamic acid (PAA) on the substrate surface, and then thermally baking it to form a self-supporting gelled film. Thermal baking was used to convert the gelled film into polyimide film. Typically, aromatic dianhydrides and aromatic diamines were polycondensed in dipolar aprotic solvent at, low temperatures, to produce PAA solution. PAA can be converted into polyimide, using either thermal imidization or chemical imidization. In this section, the chemistry, preparation, and applications of the advanced polyimide films will be described.

Aromatic polyimide films are widely used in the microelectronic manufacturing and packaging industries, particularly as the base films of flexible printed circuits (FPC), flexible packaging substrates, such as chip on film (COF), chip scale packaging (CSP), ball grid array (BGA), and so on. Line thinning of electric circuits is rapidly progressing, as a result of miniaturization and thin filming in electrical and electronic parts. Dimensional changes in polyimide films, during long-term service, can result in accidents, such as disconnection and short-circuiting, in thinner lined circuit structures. As a result, polyimide films for microelectronic applications must have high dimensional stability, a high elastic modulus, and low moisture absorption.

### 2.1. Improving Film Processability of Low-CTE Polyimide Films

Advanced polyimide films for electronic applications are typically made from aromatic dianhydrides (PMDA, or/and BPDA) and aromatic diamines (ODA, or/and PDA), among other things. Their combined properties can be tuned by adjusting the aromatic dianhydride and diamine ratios in the formation of PAAs [31]. Among the polyimide films, the PI-(BPDA/PDA) film has the highest modulus, highest *T*_g_, and lowest CTE (Coefficient of Thermal Expansion) value, owing to the rigid chemical structure of BPDA and PDA. ODA, on the other hand, has a relatively flexible ether linkage due to the possibility of the molecular chain bending and rotating. The elastic modulus increases from 3.8 to 7.8 GPa, when the mole ratio of [PDA]/[PDA+ODA] in the copolyimide PI-(BPDA/PDA-ODA) films increase, and *T*_g_ increases from 290 to 420 °C, while CTE decreases from 36 to 15 × 10^−6^ °C^−1^. It was discovered that even if only a small amount of ODA is incorporated into the polyimide backbone, the structure is disrupted and an obvious drop in *T*_g_ occurs, indicating that a simple component additivity could cause obvious changes in *T*_g_ and that the effect of the component would be sensible for rigidity.

Furthermore, because PMDA is more rigid than BPDA, the polyimide PI-(PMDA/PDA) film was expected to have a higher modulus and *T*_g_ than the polyimide PI-(BPDA/PDA). Unfortunately, the mechanical properties of the polyimide PI-(PMDA/PDA) film cannot be measured because it is too brittle. The thermal and mechanical properties of the four-component copolyimide films, derived from BPDA-PMDA and PDA-ODA, were complexly related to the polymer backbone structures. The co-polyimide film, derived from BPDA-PMDA (50/50) and PDA-ODA (70/30), demonstrated the best combination of properties, with CTE of 19 × 10^−6^ °C^−1^, elongation at breakage of 60%, modulus of 5.0 GPa, and *T*_g_ of 360 °C, respectively. To some extent, the balanced properties of polyimide films can be controlled by adjusting the polyimide backbone structures.

Z. H. Wang investigated the in-plane orientation and combined properties of the chemically imidized polyimide films [32]. A two-stage film formation process is used to create a series of fully imidized TIPI and CIPI films: the solvent evaporation stage (120 °C) and the high-temperature imidization stage (300–400 °C) (Figure 1). Variable-temperature FTIR spectroscopy, from 40 °C to 400 °C, is used to monitor the chemical and thermal imidization processes. At the solvent evaporation stage, the imidization degree, in-plane orientation, and residual solvents of films are measured. The thermal and mechanical properties of chemically imidized polyimide (CIPI) films and thermally imidized polyimide (TIPI) films were thoroughly investigated. The experimental results show that CIPI films have significantly higher tensile strength and modulus, while having a significantly lower CTE than TIPI films. These benefits are due to the high in-plane orientation and close packing of CIPI backbones. In contrast to thermal imidization, which begins at around 140 °C, chemical imidization, activated by acetic anhydride and isoquinoline, begins cyclization at room temperature. The resulting imide rings restrict polymer chain mobility and lead to in-plane orientation with solvent evaporation. Furthermore, after chemical imidization at 120 °C, fewer small molecules remain in the films than after thermal imidization.

Figure 1 depicts a comparison of ID (Imidization Degree) values as a function of temperature. Even at room temperature, the chemically imidized film has an ID of about 35%, which does not change until the temperature rises to 100 °C, indicating that the cyclization reaction can be activated by acetic anhydride and isoquinoline. From 100 °C to 200 °C, ID increases from 35% to about 90%, with only a 10% increase at higher temperatures (>200 °C). Thermally imidized films, on the other hand, show no imidization below 140 °C. According to previous research [33,34], their IDs increase noticeably, as the temperature rises from 140 °C to 250 °C. Depolymerization, orientation, plasticization, and aggregation of completely imidized PI films will all be affected by the increased ID at low temperatures for chemical imidization.

Figure 2a depicts the real-time birefringence of partially imidized films, produced by two different imidization methods during the solvent evaporation stage. Because the initial PAA films are solvent-rich, the relaxation time is reduced and the in-plane and out-of-plane refractive index values remain very close, indicating that the birefringence is close to zero. Solvents volatilize rapidly, as the standing time at 120 °C increases, and birefringence increases simultaneously, in both chemical and thermal imidizations. The IDs of TIPA-50 and CIPA-50, as a function of time at 120 °C, are depicted in Figure 2b. When the standing time at 120 °C reaches 60 min, the ID of CIPA-50 film exceeds 60%. The resulting imide rings exhibit greater intrinsic birefringence than PAA chains.

Small molecule evaporation has been shown to plastify polymer chains and disrupt the ordered arrangement of PI films at high temperatures [35,36]. Figure 3 depicts the weight loss of PA-50 films, treated for 1 h at 120 °C, via thermal and chemical imidizations, respectively. After the solvent evaporation stage, the CIPA-50 film contains fewer residual small molecules than the TIPA-50 film. At the solvent evaporation stage, the ID of the CIPA-50 film exceeds 60%, as shown in Figure 2b. Unlike PAA chains, imide rings cannot form hydrogen bonds with solvents, as illustrated in Figure 3 (inset picture).

Figure 4 shows how chemical and thermal imidizations affect the mechanical characteristics of completely imidized PI films. With increased elastic modulus and tensile strength, the CIPI films have improved mechanical properties. CIPI-50, for example, has a tensile modulus of 4.5 GPa and a tensile strength of 160.4 MPa, which is 55.2% and 50.0% higher than TIPI-50. The aggregation phase structures of the PI films are responsible for the obvious improvement in mechanical properties for CIPI films. Chemical imidization causes a higher ID and in-plane orientation degree during the solvent evaporation stage, as previously stated. As a result, the CIPI films have a higher degree of in-plane orientation and a more organized molecular aggregation phase, resulting in higher tensile strength and modulus. The enhanced stability of imide rings, generated during the solvent evaporation stage of chemical imidization, may account for the increased toughness. In the subsequent imidization process, the inflexible PMDA-PPD segments and low molecular fragments cause film toughness to deteriorate and stress cracking to occur.

The wide-angle X-ray diffraction (WAXD) patterns of the PI films were measured. All PI films made by chemical or thermal imidizations have a broad peak, about 19°, showing that they are all amorphous. The matching *d*-spacing was calculated using Bragg’s equation, based on the scattering angles (2*θ*) at the center of the broad peaks (Figure 5). In linear polymers, the *d*-spacing represents the average distance between polymer chain segments and also indicates the free volume. It is commonly used to determine the packing density of amorphous polymer films [37,38]. For thermal imidized PI films, the *d*-spacing varies from 4.25 Å to 4.72 Å and improves with increasing BAPP fraction in the polymer backbone. Chemically imidized PI films exhibit a similar trend. The inclusion of flexible BAPP segments, with a large free volume fraction in the polyimide backbone, will disrupt the regular ordered structures and limit the formation of the in-plane orientation of polymer chains, as seen by the *d*-spacing values.

The difference between the in-plane and out-of-plane refractive indices, known as birefringence, has been utilized to define in-plane molecule orientation. Figure 6 shows how the chemical and thermal imidizations affect the birefringence of completely imidized PI films. Chemical imidization can clearly yield PI films, with significantly better in-plane orientation than thermal imidization, especially for PI films with flexible segment concentrations more than 40 mol%.

Good dimensional stability is essential for the use of PI films in the fabrication of flexible electronic substrates. TMA investigated the dimension variations of PI films as a function of temperature (Figure 7). Regardless of chemical or thermal imidizations, CTE values rise as the flexible segment content in PI backbones increases. With increasing the contents of the flexible region from 30% to 100%, the TIPI films display CTE values of (21.7–63.7) × 10^−6^ °C^−1^. The PI films, on the other hand, cannot be manufactured if the flexible segment percentage is less than 30% because the films are too brittle. As a result, utilizing the thermal imidization process, obtaining PI films with CTE less than 20 × 10^−6^ °C^−1^, while maintaining considerable toughness, is difficult. At the same degree of flexible segment content in the PI backbones, the CIPI films have lower CTE values than the thermally imidized counterparts. For example, the CTE of the CIPI-50 film is 9.8 × 10^−6^ °C^−1^, which is significantly lower than that of the TIPI (CTE = 44.9 × 10^−6^ °C^−1^). The increased in-plane orientation favored by the more rigid imide rings, generated during the solvent evaporation stage, is clearly responsible for the improved dimensional stability of the CIPI films.

The physical and chemical alterations of the polymer chains, during the complete chemical and thermal processes, were proposed, in order to visibly and comprehensively characterize the effects of two imidization procedures on the structure and properties of fully imidized PI films (Figure 8). Chemical imidization, which is activated by acetic anhydride and isoquinoline, initiates cyclization even at room temperature, unlike thermal imidization, which starts at around 140 °C. The orientation, plasticization, and depolymerization of completely imidized PI films will all be affected by the high ID of the partially imidized film, achieved during the solvent evaporation stage. All of these aspects will contribute to fully imidized CIPI films having a high in-plane orientation and close packing.

Yuan et al. [39] prepared a series of molecular-weight-controlled polyimide precursors (PI-PEPA), with designed calculated molecular weight (Calc’d *M_w_*), from the reaction of BPDA and PDA, using PEPA as a chain-extendable and end-capping reagent, at concentrations of 20% in DMAc, to determine the influence of PAA molecular weights on their solution-cast processing performance (Figure 2). Termination main-chains, with the calculated molar fractions of PEPA, were used to regulate the molecular weights, yielding PAAs with Calc’d *M_w_* of 5 × 10^3^ g·moL^−1^ to 50 × 10^3^ g·moL^−1^. For comparison, an analogous series of unreactive phthalic-end-capped PAA resins (PAA-PA) was synthesized with equal regulated *M_w_* of 5 × 10^3^ g·moL^−1^−50 × 10^3^ g·moL^−1^.

For the large-scale manufacturing of high-quality polyimide films, a reasonable apparent viscosity of PAA resin with high concentration is an important solution-cast processing characteristic. Figure 9 shows the molecular-weight-controlled PAAs with a controlled *M_w_* of 25 × 10^3^ g·moL^−1^ and the molecular-weight-uncontrolled PAAs’ solution viscosity, as a function of concentration. With increasing concentration, a systematic rising trend in apparent viscosities is observed, and two regions with very different slopes suggest critical concentrations (c* and c*′) of about 10% for PAAs (BPDA-PDA) without molecular weight limitation and 20% for molecular-weight-controlled PAAs (PAA-PEPA-25 and PAA-PA-25), respectively. When the concentration is less than 20%, the apparent viscosities of PAA-PEPA-25 and PAA-PA-25 are no more than 82 Pa·s, after which, an abrupt growth is noticed by a lesser exponential rate, proportionate to the PI precursor concentration. Figure 10 shows the apparent viscosity of molecular-weight-regulated PAA resins, with a concentration of 20% in DMAc at 25 °C, as a function of molecular weight. When PAA Calc’d *M_w_* ≤ 30 × 10^3^ g·moL^−1^, the apparent viscosities of both PAA-PEPA and PAA-PA gradually grow to 120 Pa·s, following which, the seemingly exponential growths to 1160 Pa·s are depicted, advancing in decreasing order: PAA-PA > PAA-PEPA.

The mechanical characteristics of PI-PEPA and PI-PA films are compared in Figure 11, as a function of PAA Calc’d *M_w_*. The tensile strength of PI-PEPA films increases dramatically when Calc’d *M_w_* rises over 20 × 10^3^ g·moL^−1^ (Figure 11a), then appears to plateau at 237.6 MPa as Calc’d *M_w_* rises above 20 × 10^3^ g·moL^−1^, implying a critical Calc’d *M_w_* of PAA-PEPA around 20 × 10^3^ g·moL^−1^. PI-PA films, on the other hand, have much lower strength at the same Calc’d *M_w_* values. As the theoretical PAA-PEPA *M_w_* grows to 20 × 10^3^ g·moL^−1^, the elastic modulus of PI-PEPA reduces somewhat, from 7.7 to 7.3 GPa, likely due to lower chain crosslinking density (Figure 11b); the elongations at break indicate a constant increase, with PAA Calc’d *M_w_* rising. Furthermore, PI-PEPA samples exhibited longer elongations than PI-PA films.

The impact of thermal-curing methods on the crystallization behavior and morphological aspects of PI-PEPA-25 and PI-PA-25 films via WAXD is depicted in Figure 12. Both batches of films treated at lower temperatures (300 °C) show broad amorphous halos. With increasing the final curing temperature from 370 °C to 400 °C, the relative intensities of three distinguishable diffraction peaks, superimposed in the range of 18–26°, increase dramatically, and each peak’s half-width decreases, revealing a rapid enhancement in the regularity of intramolecular arrangements in polyimide films. Interestingly, when the crystalline peaks in the PI-PEPA-25 film were thermally processed at 450 °C, they became obtuse and less-structured, indicating a reduction in chain ordering, but the counterpart in the PI-PA-25 film became sharper. Figure 12c compares the effect of different curing temperatures on the ratios of crystalline areas, determined from the diffractogram of the resulting PI films. The degrees of crystallinity in the PI-PEPA-25 film rise linearly, from 7.22% to 27.26%, as the final heating temperature is increased from 350 °C to 425 °C, showing that the curing reaction occurring in the phenylethynyl moiety improves the ability to crystallize.

The effects of curing treatments on molecular orientation and thermal dimension stability of polyimide films (PAA-PEPA-25 and PAA-PA-25) are shown in Figure 13. Both films with BPDA-PDA-based backbone architectures demonstrated substantial persistence lengths in the chain direction and low chain flexibility, as well as high birefringence (Δ*n*) and low CTE, which are typically associated with a high level of in-plane chain orientation. Figure 13a shows that the uncured polyimide film with reactive phenylethynyl end-caping groups has lower birefringence and a much higher CTE than the linear control film with unreactive phthalic end-capping groups, indicating that the phenylethynyl moiety has a significant influence on chain configuration via rotational hindrance. When PI-PEPA-25 was thermally treated at <370 °C, the birefringence associated with optical anisotropy rose by 33.7%, compared to 15.9% for the control PI-PA-25 film. Figure 13b shows that when the ultimate curing temperature is raised from 350 °C to 400 °C, the CTE values of the thermally cured PAA-PEPA-25 films drop drastically from 8.5 to 5.1 × 10^−6^ °C^−1^, but those of the PI-PA-25 film increases.

The effects of the curing treatment on the mechanical properties of PI film were studied in Figure 14, to better understand the thermal-curing effect of phenylethynyl moieties in the comparatively high-molecular-weight PI film. For phenylethynyl-end-capped PAA resin (e.g., PAA-PEPA-25) changing to completely cured polyimide film, there is an ideal curing procedure (CP3), with an ultimate temperature as high as 400 °C (PI-PEPA-25). Advancement of the final curing temperature to 400 °C increases tensile strength and elongation at break by 55% and 83%, respectively, attaining the greatest results (247.1 MPa and 8.6%). After this, they plummet to 186.7 MPa and 3.4%, indicating that the failure mode has shifted to brittle fracture (Figure 14a,b). The linear PI-PA-25 shows a similar tendency, with the highest strength and elongation of the material heating up to 370 °C being 25% and 26% lower than PI-PEPA-25.

As a whole, the molecular-weight-controlled PAA resins, end-capped by reactive phenylethynyl groups with calculated *M_w_* > 20 × 10^3^ g·moL^1^, demonstrated moderate viscosities and high solid concentrations, indicating enhanced wetting/spreading ability to form continuous PAA films. The PAAs are successfully converted to fully cured polyimide films after thermal curing at 400 °C/1 h, which have better mechanical properties and thermal resistances than the high-molecular-weight, unreactive phthalic, end-capped counterpart, making them suitable for high-temperature microelectronic packaging applications.

### 2.2. Low-CTE Polyimide Films

In the film plane direction, aromatic polyimide films typically have substantially higher linear CTEs ((35–50) × 10^−6^ °C^−1^) than metal substrates (e.g., (17–18) × 10^−6^ °C^−1^ for copper foil). The polyimide/metal laminates incur thermal stress due to the CTE mismatch during the cooling process, from cure temperature to room temperature, when polyimide films were produced on a metal substrate via thermal imidization, after the solution casting of PAA solution. As a result, major issues, such as the curling, cracking, and detaching of polyimide films arise. Significant efforts have been undertaken in recent years to try to reduce the film’s CTEs. A thermal imidization-induced in-plane chain orientation phenomenon is intimately linked to the low CTE properties.

Aside from CTE, another significant factor to consider for FPC applications is the coefficient of humidity expansion (CHE). Flexible copper clad laminates (FCCL, adhesive-free polyimide film/Cu laminates) are often made by directly casting PAA solution onto copper foil. CHE stands for dimensional stability against water absorption. Polyimide films absorb water in air more readily than polyester films, owing to the highly polarizable imide groups in the backbone structures. As a result, polyimide films, having ester-linkages in their backbone architectures, have been proposed as a means of lowering water absorption.

#### 2.2.1. Ester-Bridged Low-CTE Polyimide Films

Hasegawa et al. [40] methodically examined the structure–property relationship of polyimide films containing ester-linkages in the polymer backbones (poly(ester-imide)s, PEsI). Bis(trimellitic acid anhydride)phenyl ester (TAHQ), a representative ester-containing dianhydride monomer, was chosen to react with various stiff/linear structure aromatic diamine monomers (such as, PDA, *trans*-1,4-cyclohexanediamine (CHDA), 2,2-bis(trifluoromethyl) benzi-dine(TFMB), 4-aminophenyl-4-aminophenylbenzoate (APAB), and 4,4-daminobenzanilide (DABA)), to produce PEsI precursors (PEsAA) solution, which was then poured onto the substrate to form PEsAA films, followed by thermally imidizing at high temperatures to produce PEsI films.

The PEsI films had extraordinarily low linear CTE values (3.2 × 10^−6^ °C^−1^ for PI-(TAHQ/PDA) and 3.3 × 10^−6^ °C^−1^ for PI-(TAHQ/APAB)), according to the experimental data. The para-ester connections function like a rod-like segment, allowing for in-plane orientation, generated by thermal imidization. Copolymerization with flexible 4,4′-oxydianiline (ODA) allowed for accurate CTE matching between PEsI and copper substrate, as well as a considerable increase in film toughness (Figure 15). As a result, it has been proven that introducing ester linkages into the polyimide backbone is a practical technique to manufacture polyimide films, suited for use as PFC substrates.

With favorable polymerizability between TAHQ and aromatic diamines, the reduced viscosities (*η*_red_) of the PEsAAs, produced by polycondensation of TAHQ and diamines in DMAc or NMP, ranged from 1.10 to 5.19 dL·g^−1^, suggesting that the PEsAAs exhibit high molecular weights. As a result, TAHQ has a sufficient level of reactivity. PI-(TAHQ-PDA), PI-(TAHQ-CHDA), PI-(TAHQ-TFMB), and PI-(TAHQ-ODA) are a sequence of TAHQ-derived homo-PEsI films. The PI-(TAHQ-PDA) film is a clear, high-quality yellow film. A 180 °C folding test revealed no cracks, indicating good flexibility. There was no discernible glass transition temperature up to 450 °C, indicating exceptional dimensional stability. The CTE of the PI-(TAHQ/PDA) film was equivalent to that of a silicon wafer (3.2 × 10^−6^ °C^−1^), and it showed a much higher tensile modulus (*E*) of 8.86 GPa and strength (224 MPa). The film, meanwhile, had a high birefringence (Δ*n* = 0.219), which corresponded to a high in-plane orientation. The elongation at breakage (*E*_b_) standing for toughness was not substantial enough (*E*_b_ = 5.4%), and water absorption was 1.6%. PI-(TAHQ/TFMB) possessed similar good features to PI-(TAHQ/PDA), including no glass transition, improved toughness, high tensile modulus and strength, and minimal water absorption. However, the CTE, on the other hand, was raised to 31.5 × 10^−6^ °C^−1^.

The PI-(TAHQ/ODA) film, however, exhibited the same features as the flexible-chain PI films, specifically, a high CTE of 51.2 × 10^−6^ °C^−1^ and a low *E* of 2.9 GPa. In comparison to other ODA-derived films, the *T*_g_ was surprisingly high. Three backbone structures of ODA-based films are shown in Figure 15. The *T*_g_ of the PI-(TAHQ/ODA) film values 320 °C, which is substantially greater than the *T*_g_ of the ether-linked PI-(HQDA/ODA) film (*T*_g_ = 245 °C) and PI-(BPDA/ODA) film (*T*_g_ = 245 °C) [41], indicating that internal rotations are repressed in the ester-linked polymer backbone structures. In comparison to PI-(TAHQ/PDA), the PI-(TAHQ/ODA) film has a quite high toughness (*E*_b_ = 67.2%) (Table 1). Due to the substitution of imide groups with ester groups in the polymer backbone structures, the PI-(TAHQ/PDA) film demonstrated significantly reduced water absorption (1.6%) than the traditional Kapton H film (2.5%). The decreased water absorption (0.6%) of PI-(TAHQ/ODA) film is equivalent to that of the fluorinated PI-(TAHQ/TFMB) film (0.7%).

Moreover, APAB, an ester-containing aromatic diamine, was employed to react with the stiff/linear aromatic dianhydrides PMDA, BPDA, and TAHQ, to form the PEsI films (Table 2). The lowered viscosities of the APAB-derived PEsI films ranged from 1.09 to 2.81 dL·g^−1^, demonstrating that APAB has a considerable polymerization reactivity. The cast films were all extremely flexible; Table 2 lists the parameters of PEsI films derived from APAB. In the WAXD pattern, there was a prominent and strong reflection, peaking at 22.0°, indicating the presence of a semicrystalline morphology. As explained previously, the most significant effect for imidization-induced in-plane orientation is chain linearity/rigidity (low CTE generation). The tensile modulus (*E*) and CTE of PI-(PMDA/4,4′-ODA) film (*E* = 3.0 GPa, CTE = 41 × 10^−6^ °C^−1^) and PI-(PMDA/3,4′-ODA) film (*E* = 5.0 GPa, CTE = 33 × 10^−6^ °C^−1^) [42] visually illustrate this phenomenon. The chain vectors (arrows) and probable conformations of numerous polyimide films are compared in Figure 16. At the ether linkage, the former’s chain segment is significantly bent. The latter’s chain vector is also bent at the ether linkage, but its linearity is somewhat regained by adopting a crankshaft-like shape.

To obtain PEsI films with the desired balanced properties, ODA was copolymerized with stiff/linear aromatic dianhydrides and diamines, resulting in three copolyimide films: PI-(TAHQ/PDA-ODA (70/30)), PI-(PMDA/APAB-ODA (60/40)), and PI-(TAHQ/APAB-ODA (70/30)). The characteristics of the three copolyimide films are compared in Table 3. The combined properties of the PI-(TAHQ/PDA-ODA (70/30)) film are remarkable, with a very low CTE (11.7 × 10^−6^ °C^−1^), a low water absorption (1.0%), and a high toughness (*E*_b_ = 35.0%). The incorporation of the flexible portion into the PI-(TAHQ/APAB) film is a viable option for achieving the desired combined properties. For applications of high-temperature base film in FPC boards, the film performances of a series of PEsI films were examined. The ester-containing monomers (TAHQ and APAB series) were all extremely reactive, resulting in PEsAAs with high inherent viscosities, ranging from 1.09 to 9.33 dL·g^−1^. The substituents, particularly the methoxy group, helped to reduce water absorption without compromising other desirable properties. A flexible diamine (ODA) was commonly copolymerized into highly esterified stiff PEsI films for practical FPC use. One of the PEsI copolymer films possessed excellent combined performance, including a low CTE (17.8 × 10^−6^ °C^−1^) that was nearly identical to that of copper foil as a conductive layer, with low water absorption (0.47% (*w*)), a high *T*_g_ (363 °C), and enhanced toughness (*E*_b_ > 40%).

PEsI films with stiff/linear backbone structures and lower CTE values than copper foil (17.7 × 10^−6^ °C^−1^) were prepared, using a simple copolymerization approach, with adequate amounts of flexible monomers to achieve CTE matching in flexible copper-clad laminates (FCCLs), while also drastically improving film toughness (Figure 17) [43]. Copolymerization with 4,4′-ODA significantly improved the properties of the PI-(TAHQ/M-BPTP) film, as evidenced by a significantly increased *E*_b_ (50.7%), and a substantially decreased, *W*_a_ (0.35%), as well as a managed CTE (17.7 × 10^−6^ °C^−1^), though the accompanying decrease in the tensile modulus (from 7.74 GPa to 6.35 GPa) was not obvious.

#### 2.2.2. Fluorinated Low-CTE Polyimide Films

W. Chen et al. developed a valid approach for minimizing moisture uptake of low-CTE polyimide films, by incorporating CF_3_ groups to the ester-containing aromatic polyimide backbone [44]. Bis(2-trifluoromethyl-4-aminophenyl)terephthalate (CF_3_-BPTP), a synthetic fluorinated ester-bridged aromatic diamine, was used to obtain a variety of fluorinated ester-containing polyimide films, with regulated ester-bridged segments and fluorine components in the polymer backbone (Figure 3). The PI films were synthesized by copolymerizing BPDA as an aromatic dianhydride monomer, with a mixture of PDA and various amounts of CF_3_-BPTP as aromatic diamine monomers. Measurements showed that increasing the CF3 groups loadings in the ester-containing polyimide backbones decreased the films’ water uptakes (*W*_a_), while maintaining the films’ CTEs at a low enough level. Polyimide films with targeted thermal, mechanical, and dielectric properties, for use in high-density and thinner FPCs, can be produced by optimizing CF_3_ group loadings. As a result, fluorinated ester-containing polyimide films, with a CTE of ≤20 × 10^−6^ °C^−1^ at 50–200 °C, a glass transition temperature (*T*_g_) of ≥310 °C, a Young’s modulus of ≥6.0 GPa, a *W*_a_ of ≤1.2%, and a dielectric constant (*ε*) of 3.4, were obtained. Due to their comparable CTE values, the two-layer flexible copper-clad laminate (2L-FCCL), formed by casting the polyimide precursor resin solution on the surface of copper foil and thermally imidizing at elevated temperature did not induce noticeable curling.

The effect of the ester-containing diamine content on the PAA solution viscosities is shown in Figure 18. The viscosity of the PAA-(BPDA/PDA) solution is extremely high, making casting films challenging. The solution viscosity decreased dramatically as the fluorinated ester-containing aromatic diamine (CF_3_-BPTP) was incorporated into the polymer backbone structure of PAA-(BPDA/PDA) resin. The obtained PAA resin solution viscosity dropped from 38.18 Pa·s to 2.17 Pa·s, as the mole ratio of CF_3_-BPTP/PDA was raised from 5% to 30%. Under the same conditions, the un-fluorinated ester-containing aromatic diamine (BPTP) only reduced the PAA resin solution viscosity from 86.70 Pa·s to 41.52 Pa·s, indicating that the fluorinated ester-containing aromatic diamine was more effective than the related fluorine-free one in controlling the PAA resin solution viscosity and, thus, improving the film-casting capability.

The effect of ester-containing segment content in polyimide on film mechanical properties is shown in Figure 19. At the same film thickness and treatment method, the ester-containing polyimide films exhibited a higher Young’s modulus, in the range of 6.0–6.9 GPa, which was analogous to that of the commercial PI-(BPDA/PDA) film (6.0 GPa). This can be explained by the following two factors: (1) The polyimide backbone, which is made up of a *para*-bonded aromatic ester-containing segment and rigid BPDA-PDA skeletons, has a high degree of in-plane orientation; (2) the strong intermolecular interactions between the imide carbonyl groups and ester linkages result in a physical cross-linking effect in the polymer backbones.

The water resistance of the aromatic polyimide film for application in high-density FPC is a major concern. The humidity expansion of FPC is proportional to its water uptake. Figure 20 depicts the relationship between the ester-containing segment loadings in the polyimide backbone and the water uptakes in the film. Water uptakes in the PFEI-I series films decreased with increasing ester-containing segment loadings, from 2.6% for PFEI-I_a_ (5% (mol) of CF_3_-BPTP) to 0.7% for PFEI-I_d_ (30% (mol) of CF_3_-BPTP).

The thermal dimension stabilities and in-plane CTE of the fluorinated ester-containing polyimide films are shown in Figure 21. Small dimensional increases were observed in the polyimide films, until the scanning temperature reached the glass transition temperature, at which point the dimensional increases abruptly changed (Figure 21a). The CTE values increased linearly as the ester-containing segment loadings in the polyimide backbones were increased (Figure 21b). Due to its most linear/rigid backbone structure, the PFEI-I_0_ had the lowest CTE of 3.3 × 10^−6^ °C^−1^, compared to 8.3 × 10^−6^ °C^−1^ for PFEI-I_a_ (5% (mol) of CF_3_-BPTP) and 18.3 × 10^−6^ °C^−1^ for PFEI-I_d_ (30% (mol) of CF_3_-BPTP), respectively, possibly due to the large free volume and relaxed chain packing prompted by CF_3_ groups. The inclusion of fluorine-free BPTP, on the other hand, did not result in a severe dimensional increase like the fluorinated CF_3_-BPTP, whose CTE values shifted from 3.7 to 6.3 × 10^−6^ °C^−1^, as the ester-containing segments increased. As a result, the rigid structure of the para-aromatic ester-containing segments acted as another essential factor for the in-plane orientation, similar to the rod-like poly(BPDA/PDA) skeleton, resulting in decreased thermal expansion.

### 2.3. Colorless Transparent Polyimide Films for Flexible Display Applications

Aromatic polyimide films are widely utilized in the microelectronics and aerospace industries, but their yellow to brown colors limit their application in optical and display devices. The intermolecular CTC, generated as a result of molecular aggregation between polymer chains in the solid states, is principally responsible for the color of conventional aromatic polyimide films. Colorless polyimide films with good heat resistance are in high demand due to the rapid advancement of optoelectronic engineering. The processing temperature on flexible polymer film substrates may be higher than 300 °C in the manufacturing of flexible active-matrix organic light emitting display devices (AMOLEDs) [45,46]. At such a high processing temperature, most typical polymer optical films, such as polyethylene terephthalate (PET) or polyethylene naphthalate (PEN), lose their optical and mechanical characteristics. As a consequence, colorless and optically transparent polymer optical films with high-temperature resistance have stimulated the interest of academics and engineers alike.

One of the most complicated aspects of developing colorless transparent polyimide (CTPI) films is balancing thermal properties, optical transparency, and mechanical strength. The introduction of alicyclic substituents, including cyclobutane, cyclohexane, cardo groups (diphenylcyclohexane, adamantane, etc.); highly electronegative groups, such as trifluoromethyl (CF_3_); asymmetrical or twisted rigid substituents, such as asymmetrically substituted biphenyl or diphenylether moieties, are among the promising designs that could enhance both the high-temperature stability and optical transparency. These functional groups or structural segments have been commonly used to make new CTPI films, enabling in them superior thermal stability (*T*_g_ ≥ 300 °C), as well as outstanding optical transmittance (>85% in the visible light spectrum). At the same time, these substituents or structural segments may prevent the development of CTC in CTPI films caused by electron-donating and electron-withdrawing moieties.

Fluorinated polyimides synthesized from 6FDA (2,2-bis(3,4-dicarboxyphenyl) hexafluoropropanedianhydride) and aromatic diamines are transparent [47,48,49,50,51]. Polyimides prepared from an alicyclic diamine, DCHM (4,4′-diaminodicyclohexylmethane), and aromatic dianhydrides are also transparent and thermally stable. As a result, semi-alicyclic polyimides have been thoroughly investigated, in order to obtain colorless transparent polyimide films. The fluorescence spectra and ultraviolet-visible absorption spectra of PI (6FDA/DCHM) and PI (6FDA/PDA) films were examined [52]. In the region of the visible spectrum, both films displayed high transparency, with no absorption bands above 370 nm. The bulky group -C(CF_3_)_2_- precludes molecular packing in polyimide films, resulting in weak intermolecular interactions between the diamine and diimide moiety as electronic donors and acceptors. Because of the steric obstacle, introducing 6FDA into polyimide chains could reduce intermolecular charge transfer [53]. The poor electron-donating capability of DCHM, on the other hand, would restrict not only intermolecular but also intramolecular charge transfer. An extremely transparent polyimide film was obtained by introducing 6FDA and DCHM into the polyimide chain. Both films exhibit *T*_g_s of around 250 °C, demonstrating that the alicyclic groups did not impair the thermal stability of polyimides.

A semi-aromatic polyimide film with a high transparency and colorlessness was prepared using alicyclic dianhydride and aromatic diamine at ICCAS (Figure 4) [54]. 2,2′-bis[4-(3-amino-5-trifluoromethylphenoxy) phenyl]sulfone (*m*-6FBAPS), a unique *meta*-substituted aromatic diamine with trifluoromethyl and sulfonyl groups in the backbone, was synthesized and utilized to react with 1,2,4,5-cyclohexanetetracarboxylic dianhydride (CHDA), to form a semi-aromatic polyimide film. A number of additional semi-aromatic polyimides were produced by employing CHDA and other different aromatic diamines for comparison.

The optical transparency of semi-aromatic polyimide films is shown in Figure 22, as measured by their physical appearance and UV-vis spectra. Even though the film thickness was as high as 70 μm, these films were transparent and basically colorless, in contrast to the deep-yellow or brown color of conventional aromatic polyimide films. The alicyclic moieties in the polymer structure are responsible for the increased optical transparency. Table 4 summarizes the optical transparency obtained from UV-vis spectra, including cutoff wavelength (absorption edge) and transmittance at various wavelengths. All of the semi-aromatic polyimide films demonstrated exceptional optical transparency, with UV cutoff wavelengths less than 314 nm and transmittance at 450 nm, greater than 91%. In contrast, the typical aromatic polyimide film derived from PMDA and 4,4′-ODA (ref-PIb) exhibited a deep-yellow color and a 2% transmittance at 450 nm.

Polyimides, including bulky electron-withdrawing trifluoromethyl and sulfonyl groups in diamine moieties, such as PI-2, PI-5, and PI-6, formed highly transparent and colorless films, with T_400_ of 90% and T_450_ of 94%, respectively. The deformed molecular conformation, combined with the impaired electron-accepting and electron-donating capabilities of dianhydride and diamines, which considerably restricted the generation of inter-/intramolecular charge transfer interactions, are credited with their good optical features.

Furthermore, a high-temperature radio-frequency magnetron sputtering method for producing transparent and conductive indium tin oxide (ITO)/polyimide films has been designed at ICCAS (Figure 23). ITO/PI films were fabricated via radio-frequency magnetron sputtering at an increased substrate temperature, using a highly transparent and thermally stable polyimide (Figure 24). To manufacture the extremely transparent and conductive ITO/PI films, a two-step deposition approach was established, in which a thick bulk ITO layer was overlapped by deposition on a thin seed ITO layer, with a dense surface. Physical appearances, transmittance intensities were used to evaluate the optical properties of the ITO-m/PI and ITO-b/PI films. Both films were exceptionally transparent and colorless, with average transmittances of over 81% in the 400–800 nm range and *b** values of less than 3 (Table 5). ITO-b/PI showed a resistivity of 9.03 × 10^−4^ Ω·cm, which was two orders of magnitude lower than ITO-m/PI. After annealing at 240 °C, the ITO/PI film possessed an 83% transmittance and a sheet resistance of 19.7 Ω·square^−1^.

A method for manufacturing colorless and transparent PI films, using a solution casting procedure, was reported by Mitsubishi Gas Chemical Company. The bi-axially stretched colorless PI films have high optical transparency, heat resistance, and low dimension changes [55]. The films were made using soluble CTPI resin, which was synthesized from CBDA and aromatic diamines, via a one-step high-temperature polycondensation process. The resulting CTPI film had a thickness of 200 µm, a total light transmittance of 89.8%, a yellow index of 1.9, and a haze of 0.74%. The film had a 0.5% solvent residual ratio by weight. Colorless CTPI films could have a wide range of applications in optoelectronics, including transparent conductive film, transparent substrates for flexible displays, flexible solar cells, and flexible printed circuit boards (FPCB), because of their unique optical and thermal features. Similar approaches have been reported in the literature [56].

In addition, DuPont Company [57] and Kolon Industries [58] revealed the ways to form colorless transparent polyimide films from fluoro-containing dianhydride copolymers. Flexible and robust CTPI films, with achromatic color and high transparency, were obtained. The Langley Research Center of NASA (National Aeronautics and Space Administration, USA) has researched molecularly orientated transparent polyimide films for space applications [59]. A colorless and transparent polyimide film, LaRC-CP1, is prepared using soluble polyimide resin with 6FDA and fluoro-containing diamine. After stretching treatment, the tensile strength of 2.0× stretched film increased from 93.0 MPa to 145.4 MPa, and the elongation increased from 16 to 65%. The dimensional stability, toughness, elongation, and strength of the films were considerably improved after stretching, which is critical for applications in space conditions.

### 2.4. Inorganic Hybrid Polyimide Films

Hazardous conditions, mainly atomic oxygen (AO) and vacuum ultraviolet (VUV), can degrade polymer materials used on the exterior surfaces of a spacecraft in low Earth orbit (LEO). Typically, multi-layer thermal insulation (MLI) is used to protect the outside surfaces of a spacecraft, with the outer surface of MLI consisting of exposed polyimide films, such as Kapton H, Upilex-R, and others. The MLI performances of these polymer films are always deteriorated as a result of AO and UV-induced erosion. Optically transparent inorganic thin films have traditionally been used as overlay protective coatings to protect polyimide films from AO bombardment [60]. The existing hard coating technology, on the other hand, has three flaws. Pinholes in protective coatings produced during the coating process are one example. Even small pinholes have been proven to cause significant undercutting in polymer layers [61]. As a result, the number of pinholes or imperfections in the inorganic coating must be kept to a minimum, in order for the polymer films to survive. For existing thin-film techniques, this is a major technological barrier. Another issue is the formation of cracks in regions where cutting, punching, or a sharp bend has occurred. The cracks are larger than pinholes, which are the most common cause of MLI damage. Last but not least, collisions with space debris or micrometeoroids in orbit may cause gaps in the inorganic coatings, allowing AO to react with the polymer beneath. Consequently, spacecraft protective coatings with self-healing capabilities are desired in LEO environment.

Kapton polyimide film is widely applied in solar arrays, spaceship thermal blankets, and space inflatable constructions. The film is extensively damaged when exposed to AO in LEO. Incorporating polyhedral oligomeric silsesquioxane (POSS) into the polyimide backbone frameworks by copolymerizing a monomer mixture of POSS-containing diamine and other aromatic diamine with aromatic dianhydride in a polar solvent to produce a high-molecular-weight PAA solution, is then cast on a substrate surface to form a PAA film [62]. A self-standing gel film is peeled off from the substrate surface, followed by imidization to obtain a POSS-Kapton polyimide film, after some of the solvent is removed by evaporation and partly chemical or thermal imidization (Figure 5). The POSS is a remarkable class of nanoscale inorganic/organic hybrid cage-like architectures, with a silicon/oxygen framework (RSiO_1.5_), where R is a hydrocarbon group used to tune polymer compatibility and chemical stability. On the Materials International Space Station Experiment (MISSE) platform, POSS-Kapton films were subjected to AO in space. AO flux and fluence differed from experiment to experiment, based on exposure time, experiment period within the solar cycle, and MISSE sample tray orientation. The AO’s relative velocity to the ram surface is about 7.4 km·s^−1^, equating to oxygen atoms impacting the surface at a velocity of 4.5 eV on average. After 3.9 years, the MISSE-1 flight experiment was recovered. The samples were exposed to the ram and, hence, to all constituents of the LEO environment, including AO and VUV irradiation, with the O-atom fluence being approximately 8 × 10^21^ atoms·cm^−2^, comparable to the fluence on the companion MISSE-2, which was accurately determined at 8.43 × 10^21^ atoms·cm^−2^.

At ambient temperature, the SC-POSS Kapton polyimide film was subjected to AO, with a total fluence of around 2.7 × 10^20^ atoms·cm^−2^. The laboratory AO erosion measurements of SC-Kapton film are included in Table 6 [62]. The experimental results showed that increasing the content of SC-POSS reduced surface erosion dramatically. The atomic-oxygen erosion yields of SC POSS-Kapton and MC POSS-Kapton films were comparable.

Temperature effects on the atomic-oxygen erosion yield of POSS-Kapton films with Si_8_O_12_ loadings of 0%, 3.5%, and 7.0% were examined. Table 7 illustrates the AO erosion depths of MC-Kapton films in the lab at various temperatures. From 25 °C to 300 °C, the erosion depth of the 0%(*w*) MC POSS-Kapton film reference increased by a factor of around 3.6, indicating the largest temperature dependence. Furthermore, the erosion of 3.5 and 7.0%(*w*) Si_8_O_12_ MC POSS-Kapton films was less temperature dependent. With the increase in temperature, from 25 °C to 300 °C, the erosion depths of these samples increased by factors of 2.2 and 2.4, respectively.

On MISSE-1, 0%(*w*), 1.75%(*w*), and 3.5%(*w*) MC POSS-Kapton films were flown in ISS for 3.9 years. Throughout the flight, several photographs of the samples were obtained, and it was observed that the 0%(*w*) MC POSS-Kapton film reference sample was entirely eroded in less than 4 months. The 0%(*w*) MC POSS-Kapton film should have eroded a minimum of 240 µm, based on an estimated O-atom fluence of 8 × 10^21^ atoms·cm^−2^ and an erosion yield of 3.00 × 10^−24^ cm^3^ per atom for standard Kapton H film. The 1.75%(*w*) and 3.5%(*w*) MC POSS-Kapton films were degraded by 5.8 µm and 2.1 µm, respectively, according to the experimental data.

Miyazaki et al. researched the AO tolerance of a polysiloxane-block polyimide film [63,64]. The silicon-containing polyimide film (BSF30), a polysiloxane-block-polyimide, was chosen as the subject of inquiry. At JAXA’s Combined Space Effects Test Facility in Tsukuba, Japan, an AO beam was bombarded on a polysiloxane-block-polyimide film. The new layer is built with a 500 nm deep damaged area, as shown by AO irradiation of the eroded surface. The ability of the polysiloxane-block-polyimide to “self-heal” was identified, implying that polysiloxane-block-polyimide film has a great deal of potential as a space-use material, especially for LEO environment.

A set of AO-resistant and transparent polyimide coatings incorporating phosphine have been designed [65]. The Williamson reaction of 3,5-difluorophenyldiphenylphosphine oxide (DFPPO) and *meta*-aminophenol yielded [3,5-bis(3-aminophenoxy) phenyl]diphenylphosphine oxide (*m*-BADPO), a *meta*-substituted aromatic diamine. Afterwards, the aromatic diamine was polymerized using a variety of commercially available aromatic dianhydrides to produce a series of aromatic polyimides (PI-1 to PI-4) (Figure 6).

The solubility of the polyimides was found to be improved due to the synergistic effects of *meta*-linked structure and bulky PPO groups, allowing for solution casting fabrication of the polyimide films. In the visible light wavelength, the films demonstrated flexible and robust features, with light color and good transparency (Figure 25a). The films exhibited up to 87% transmittance at 400 nm. At the same thickness, PI-4 had a cutoff wavelength of 319 nm, which was 7 nm lower than polyimide from *p*-BADPO and 6FDA. The PI-4 film’s transmittance at 400 nm (85%) was also higher than its *para*-linked counterpart (82%). The yellow index (YI) is a commonly used criterion for assessing the coloration of a polymer film. This value describes how even a film sample’s color shifts from clear or white to yellow. A lower YI value generally suggests a polymer film with poor coloring. The loose molecular packing, driven by the bulky pendant PPO group in the diamine monomer, and the flexible ether or 6F atoms in the aromatic dianhydride in PI-3 and PI-4 films, could be attributed to their good transparency and low coloring.

At the ground-based simulation facility, the polyimide films PI-1, PI-3, and PI-4, as well as the Kapton reference (20 × 20 × 0.05 mm), were exposed to AO. As a standard, the Kapton erosion rate was set to 3.0 × 10^−24^ cm^3^·atom^−1^. The specimens were subjected to AO, with a total exposed fluence of 8.13 × 10^20^ atoms·cm^−2^. The mass loss against AO fluence for the polyimide films and the Kapton reference can be seen in Figure 25b. The polyimide samples containing the PPO group had a weight loss of 5.48–8.76%(*w*), while Kapton H showed a weight loss of 56.76%(*w*).

XPS was used to investigate the surface composition of the AO-exposed PI films. The results revealed that a phosphorus oxide layer formed on the surface of the PI-1 film. This inert layer may prevent the underlying polyimide film from eroding deeper. After AO exposure, SEM images of Kapton H and PI-1 samples are compared in Figure 26. Both the surface appearances of PI-1 and Kapton films presented a typical carpet-like pattern.

F. L. Liu et al. developed carborane-containing aromatic polyimide (CPI) films, with extraordinary thermo-oxidative stability at 700 °C, formed by casting poly(amic acid) (PAA) resin solution on a glass substrate and then thermal imidizing at high temperatures [66]. In an aprotic solvent, an aromatic dianhydride and a mixture of aromatic diamines, including carborane-containing aromatic diamine, were copolymerized to produce the PAA solution. Because of the multilayered protective layers generated on the film surface by the pyrolysis process of the carborane group into boron oxides, the CPI films demonstrated exceptional thermo-oxidative stability at 700 °C. In a high-temperature condition, the boron oxide layer increased the degradation activation energy and restricted the direct interaction of inner polymer materials with molecular oxygen, performing as a “self-healing” upmost coating layer on the polyimide materials. Even after thermo-oxidative aging at 700 °C for 5 min, the CPI-50 film remained flexible and retained 50% of its mechanical strength.

The preparation process of the CPI films is shown in Figure 7. Polycondensation of equimolar aromatic dianhydride (*s*-BPDA), with a mixture of three distinct aromatic diamines, including PDA, 4,4′-ODA, and 1,7-bis(aminophenyl)-*meta*-carborane (BACB) in NMP, afforded the polyimide precursor PAA resin solution. The researchers designed a series of CPI-*x* films with different boron loadings (*x* means the molar ratio of BACB in diamine monomers and is equivalent to 0, 5, 10, 20, 30, 40, and 50, respectively).

The thermal stabilities, as measured by DMA and TGA, are shown in Figure 27. Though the bulky carborane cage structure increased the distance between polymer chains, it also restrained polymer chain mobility in the large scale due to its large steric hindrance, and the movements prompted by the larger interchain space were too tiny in size scale to change polymers from a glassy to rubbery state, leading to higher *T*_g_ values. Incorporating carborane into the polyimide backbone structure dramatically improved polymer thermal and thermo-oxidative stabilities, as compared to the boron-free film (CPI-0) (Figure 27b,c). Increased boron loadings enhanced both thermal and thermo-oxidative stability.

Due to the strong electron-withdrawing capability and massive steric hindrance effect of the carborane cage structure, BACB is much less reactive than the other two aromatic diamines (4,4′-ODA and PDA) during copolymerization with aromatic dianhydride, limiting the polymerization reactivity of the diamine monomer mixture. BACB interacted with dianhydride to generate a lower *M_w_* polymer, despite its difficulty in forming a high *M_w_* polymer. Hence, the molecular weight of PAA precursors decreased, and the polymer interchain interaction deteriorated as boron loadings increased, resulting in reduced tensile strength and modulus. In comparison to CPI-20 (*T*_S_: 161 MPa, *T*_M_: 4.9 GPa), the CPI-50 film possessed the lowest mechanical properties, with a tensile strength (*T*_S_) of 108 MPa and a tensile modulus (*T*_M_) of 3.7 GPa.

CPI films with varied boron loadings (CPI-0, CPI-20, and CPI-50) were thermally treated under a series of thermo-oxidative aging test conditions (600 °C/30 min, 600 °C/60 min, 700 °C/5 min, 700 °C/15 min, 700 °C/30 min). Bend testing was used to determine the flexibility of the CPI films, and tensile characteristics were determined on an Instron 5567 universal testing instrument, with a drawing rate of 2.0 mm·min^−1^. Under the same circumstances, all of the aged boron-containing films could be bent 180°, demonstrating excellent flexibility retention (Figure 28a). Moreover, despite the fact that the CPI-20 film was split into two pieces during bend testing, following thermo-oxidative aging at 700 °C for 30 min, the CPI-50 film still maintained good flexibility.

After thermo-oxidative aging at 600 to 700 °C, both CPI-20 and CPI-50 films exhibited appreciable tensile strength (Figure 28b); the unaged CPI-20 and CPI-50 films possess tensile strengths of 161 MPa and 108 MPa, respectively. The tensile strength of CPI-20 and CPI-50 was degraded to 88 MPa and 79 MPa, after thermo-oxidative aging at 600 °C for 30 min. The tensile strength was degraded to 23 MPa (CPI-20) and 42 MPa (CPI-50) when the aging period was extended to 60 min at 600 °C, demonstrating that the thermo-oxidation at 600 °C increased with increasing aging duration. Furthermore, the CPI-50 film successfully retained tensile strength of 63, 35, and 18 MPa, after aging at 700 °C for 5, 15, and 30 min, and its strength retention was substantially higher than that of CPI-20 (Figure 28c). The introduction of the carborane cage into the polymer backbone structure considerably enhanced the polyimide thermo-oxidative stability at 700 °C, when compared to boron-free films.

Additionally, three representatives, with differing carborane cage loadings (CPI-0, CPI-20, and CPI-50), were isothermally treated by TGA in air at 700 °C for 150 min, for mechanism investigation (Figure 29a). Increased carborane cage loadings appear to have considerably enhanced thermo-oxidative stability. After 150 min of thermo-oxidative aging at 700 °C, the boron-free CPI film (CPI-0) was fully degraded into volatiles, while the CPI-20 and CPI-50 films still kept more than 29% and 76% weight residue, respectively. Furthermore, the boron-free CPI film (CPI-0) was easier to degrade than the carborane-containing films (CPI-20 and CPI-50), with a stronger decomposition peak. When combined with DTG curves (Figure 29b), a second fast weight loss peak for CPI-0 in air occurred, almost immediately after the first. For CPI-50 in air, however, just one decomposition peak was seen. This might be explained by the fact that, due to the larger *E*_a_ of the boron oxide passivation layer formed during the thermal processing, further decomposition of the CPI-50 films was postponed.

The thermal degradation can be separated into three stages, as shown in Figure 30. The first stage, according to the XPS measurement, was the degradation of the exterior polymer, with the surface becoming increasingly rough. The upmost film was reacted with oxygen in this stage, and volatiles, such as oxynitride, oxycarbide, and hydroxide, were released, along with the generation of a B-O bond. The second stage was the formation of a boron oxide protective barrier on the film surface. At first, the oxidative layer covered a portion of the surface (Figure 31b). Accordingly, the polymer on the upmost passivation layer next to the opening hole was then degraded in situ, forming a new passivation layer, allowing the hybrid polyimide surface to “self-heal” [67]. Fracture and regeneration of the protective layer were in a dynamic balance at this stage, but the regenerated layers progressively repaired the previous ones, eventually establishing the multilayered passivation layer (Figure 31e). This layer encased the film surface and avoided direct interaction between the underlying polymer and oxygen, resulting in a considerable reduction in the rate of decomposition. Despite the formation of a “self-healing” multilayered dense passivation layer on the film surface in the second stage, heat could still be carried into the inside via the protective covering, causing the polymer to deteriorate slowly in the final stage.

Up to 700 °C, the carborane-containing polyimide (CPI) films demonstrated extraordinarily good thermal and thermo-oxidative stability, attributed to the multilayered protective layers generated on the film surface by the pyrolysis process of the carborane cage into boron oxides. In a high-temperature condition, the boron oxide layer increased the degradation activation energy and restricted direct interaction of interior polymer materials with oxygen molecules, functioning as a “self-healing” cover coating on the polyimide materials. Even following thermo-oxidative aging at 700 °C for 30 min, the CPI films maintained substantially higher mechanical strength and flexibility than the similar carborane-free films. This “self-healing” multilayered dense boron oxide barrier delayed the further degradation of the underlying polymer and extended the life span of materials at elevated temperatures.

## 3. Conclusions and Prospects

The representative progress in aromatic polyimide films for electronic applications, mainly from our laboratory, have been reviewed, especially on the preparation, structure and properties. Aromatic polyimides have prominent thermal stability, high mechanical strength and toughness, high electric insulating properties, low dielectric constants and dissipation factors, as well as high radiation and wear resistance, etc., and have been extensively used in microelectronics and optoelectronics. However, in order to meet the new microelectronic industry demands for flexible PI films, their combined performance must then be upgraded. Future research directions for electronic applications of PI films, according to our understanding, will be focused on: (1) The precise control of chemical reactions and physical changes during film formation, especially chemical imidization and bidirectional stretching, which have a great impact on the properties of the resultant films. (2) Surface modification of polyimide films, such as plasma treatment, is also necessary in electronic applications. (3) Compared with the synthesis of novel monomers, it is often more effective to obtain better balanced properties by designing the molecular structure of copolyimide films. (4) The organic–inorganic hybrid method can easily give polyimide films unique properties, such as heat resistance and electrical properties, which are difficult to achieve for traditional organic polymers, and will gradually become the mainstream research direction in the future.

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
