# Peer review of "Progress in Aromatic Polyimide Films for Electronic Applications: Preparation, Structure and Properties"

_polymers, 2022, doi:10.3390/polym14061269_

Round 1

Reviewer 1 Report

This review summarizes progresses in the development of polyimide films for electronic applications with good overall organization. Because the contents would be valuable for polymer chemists and material scientists, the reviewer recommends this review for publication in Polymers. However, the following issues are recommended to be addressed in revision of the manuscript.

  1. The major issue the reviewer has is whether this review showcases the "recent" progress of the topic. In the abstract, the authors described that "In this review, the recent progress in aromatic polyimide films for electronic applications will be described". However, the reviewer found that throughout the entire review, the majority of the references range from 2004-2016, with some articles/reviews referenced dating back to the 1990s. While older papers and reviews are highly useful for readers to understand the context and scientific background of the topic, more recent development of the topic should be referenced and discussed to keep readers up to date of the "recent progress" of the topic. For example, Gu et al. recently (2021) demonstrated the fabrication of several novel kinds of intrinsically highly thermally conductive liquid crystalline polyimide films (Macromolecules 2021, 54, 10, 4934–4944). A recent review of the topic can also be useful for the authors: https://doi.org/10.1016/j.coco.2019.08.011. 
  2. The term "CTE" is one of the most important concepts used in this review. The definition of the term "coefficient of thermal expansion (CTE)" should be explained when the term was first introduced in the paper (page 2, line 65) instead of later in the page (page 2, line 95).
  3. While the authors provided a decent conclusion section, the reviewer suggests that the authors may potentially provide some of their insights on how the topic can further develop in the future. Adding an outlook section for this review can give future readers some ideas on how the next generation of PI films may improve. For example, Kim & Han et al. has recently (Polymers 202113(21), 3824) reported a hydrothermal synthetic method to be a “greener” and more facile method for sustainable PI synthesis. The authors should use their expertise in the field to briefly guide future researchers how the synthesis and fabrication of PI films can potentially develop.
  4. The authors employ a significant amount of abbreviations throughout the review. As a review, it would be beneficial for readers to keep track of these abbreviations if the authors could make sure all of these abbreviations are clearly explained. For example, the definition the term "ID" in the caption of Figure 1 (page 3, line 102) was not given.

Author Response

Thank you for reading this manuscript carefully and pointing out very pertinent questions,we have made appropriate revisions to the relevant problems:

1. There may be some shortcomings in this review that references and contents are not up to date. Since this is an invited review by the chief editor, the authors hopes to introduce the representative research works of the research group in detail as much as possible, so the writing method is somewhat different from other review articles.  According to your suggestion, more recent publications were attached for the reference of interested readers on the basis of retaining the original content.

2. Thank you again for reminding me. This problem has indeed not been noticed before, and has been corrected in the text.

3. Good advice. We have carefully read the references you provided, and after careful consideration, we have summarized some future prospects for this research direction.

4. The definitions of the abbreviations have been explained.

Reviewer 2 Report

I should say that there is a critical lack of literature in the main text and figures. Since this is a review paper (not an original paper), all the figure-captions cited should have references, but almost all captions lack reference. In addition, the lack of reference is also significant in the main text. I cannot follow the original articles very frequently. A totally revised manuscript should be re-submitted. Then, I will review it from the beginning.

Author Response

Thank you for reading this manuscript carefully and pointing out very pertinent questions. There is indeed a lack of references and content in the last manuscript. Since this is an invited review by the chief editor, the authors hopes to introduce the representative research works of the research group in detail as much as possible, so the writing method is somewhat different from other review articles.  According to your suggestion, more recent publications were attached for the reference of interested readers on the basis of retaining the original content, and the sources of the figures and tables are also cited.

Round 2

Reviewer 2 Report

Despite the introduction words in the Abstract: "In this review, the representative progress in aromatic polyimide films for electronic applications will be described, especially on the preparation, structure and properties", this review mainly introduce only a very limited number of papers at a great length for each, particularly those published by the authors. As obviously seen in the manuscript, almost all figures are taken from refs. 32, 39, 40, 44, 54, 62, 65, and 66, and six of them are from the authors' group, which is significantly biased. I should say that the broad title of 'Progress in Aromatic Polyimide Films for Electronic Applications' and the Abstract are misleading. This paper must be a special summary focusing on a particular area studied by the authors. The title and abstract should be revised to correctly express the purpose and status of this manuscript.

Author Response

Thank you again for your timely feedback on the problems in our manuscript. The suggestion you pointed out is indeed very accurate. Since we mainly introduce the work of our own laboratory, the previous broad title and abstract are unreasonable. We have corrected them appropriately to avoid misleading readers.

Round 3

Reviewer 2 Report

Thank you for the lots of revisions. The references for Figs.23, 24, Table 5 are still missing, but the revised manuscript can be accepted as a review article of Polymers.